# Augmenting Speech Quality Estimation in Software-Defined Networking Using Machine Learning Algorithms

**DOI:** 10.3390/s21103477

**Published:** 2021-05-17

**Authors:** Jan Rozhon, Filip Rezac, Jakub Jalowiczor, Ladislav Behan

**Affiliations:** 1Deparment of Telecommunications, Faculty of Electrical Engineering and Computer Science, VSB-Technical University of Ostrava, 17. listopadu 2172/15, 708 00 Ostrava, Czech Republic; jan.rozhon@vsb.cz; 2CESNET z.s.p.o., Zikova 4, 160 00 Prague, Czech Republic; jakub.jalowiczor@vsb.cz (J.J.); ladislav.behan@vsb.cz (L.B.)

**Keywords:** speech analysis, software defined networks, OpenFlow, artificial neural networks

## Abstract

With the increased number of Software-Defined Networking (SDN) installations, the data centers of large service providers are becoming more and more agile in terms of network performance efficiency and flexibility. While SDN is an active and obvious trend in a modern data center design, the implications and possibilities it carries for effective and efficient network management are not yet fully explored and utilized. With most of the modern Internet traffic consisting of multimedia services and media-rich content sharing, the quality of multimedia communications is at the center of attention of many companies and research groups. Since SDN-enabled switches have an inherent feature of monitoring the flow statistics in terms of packets and bytes transmitted/lost, these devices can be utilized to monitor the essential statistics of the multimedia communications, allowing the provider to act in case of network failing to deliver the required service quality. The internal packet processing in the SDN switch enables the SDN controller to fetch the statistical information of the particular packet flow using the PacketIn and Multipart messages. This information, if preprocessed properly, can be used to estimate higher layer interpretation of the link quality and thus allowing to relate the provided quality of service (QoS) to the quality of user experience (QoE). This article discusses the experimental setup that can be used to estimate the quality of speech communication based on the information provided by the SDN controller. To achieve higher accuracy of the result, latency characteristics are added based on the exploiting of the dummy packet injection into the packet stream and/or RTCP packet analysis. The results of the experiment show that this innovative approach calculates the statistics of each individual RTP stream, and thus, we obtain a method for dynamic measurement of speech quality, where when quality decreases, it is possible to respond quickly by changing routing at the network level for each individual call. To improve the quality of call measurements, a Convolutional Neural Network (CNN) was also implemented. This model is based on two standard approaches to measuring the speech quality: PESQ and E-model. However, unlike PESQ/POLQA, the CNN-based model can take delay into account, and unlike the E-model, the resulting accuracy is much higher.

## 1. Introduction

With the rise of new paradigms in data center architectures, such as Network Function Virtualization (NFV), dynamic resource sharing, and resource isolation, the need for a more flexible way to manage and allocate network resources has also arisen. To accommodate this need, several technologies that completely changed the way how modern network environments and infrastructures work have been invented and developed. The first attempts to move from the standard decentralized way of network resources management were FORCES in 2003 [1] or Routing Control Platform (RCP [2]) in 2004 that both proved immature for general acceptance and implementation. It was not until 2006 when a new communication interface for network device configuration called NETCONF was proposed by the Internet Engineering Task Force (IETF) that networks began to be designed according to this new paradigm in a noticeable scale.

The surge in the use of these technologies came with the development of OpenFlow protocol originally invented at Stanford University in 2008 and later maintained and further developed by the Open Networking Foundation (ONF). Thanks to the open approach to the protocol development, the community around OpenFlow quickly grew, and now, many of the big companies cooperate under the flag of ONF to further develop OpenFlow protocol.

OpenFlow defines a standard interface between the center of network logic—a controller, and relatively simple network switches that do not have the static configuration but are rather programmed dynamically by instructions coming from the controller. This programmatic approach to network configuration is often called Software-Defined Networking. The main features and advantages of SDN networks employing OpenFlow protocol include the following:Control and data plane separation,Centralized network management,Standardized logical structure of switch components,General and standardized interface (API) for data plane instructions installation.

Control and data plane separation is one of the key aspects of SDNs that allows for an increase of network maintenance efficiency and fast network reconfiguration with both performed in a centralized fashion. During the evolution of OpenFlow, several tables have been introduced into the OpenFlow capable switch architecture. The main components now include the following:Flow tables,Group tables,Meter tables.

Each table of an OpenFlow-capable switch contains prioritized instructions that are matched against the incoming traffic that is distinguished and separated into individual traffic flows. A traffic flow is an abstraction that encompasses all the packets of the incoming traffic that have common field values such as Internet Protocol (IP) addresses, or transport layer port numbers. Individual tables can also be stacked into more complex structures that allow for combining the instructions into instruction sets and further increase the flexibility of such a setup.

For each of the flow/group/meter table entry, the OpenFlow-capable switch also stores the information about the matching packets and their byte counts. This is done using the Counters field in the table entry. This information can be retrieved by the controller using the Multipart Request/Reply Messages (OFPT_MULTIPART_REQUEST/OFPT_MULTIPART_REPLY) as defined in OpenFlow 1.5.1 [3].

In this paper, we propose a method suitable for obtaining the information from the OpenFlow capable switch that can be used to compute the estimation of Mean Opinion Score (MOS) statistics of individual audio calls. Then, these statistics can be used to reconfigure the network in terms of a logical topology or packet handling policies to enhance the overall QoS and QoE statistics of the audio transmissions. As a result, a more robust and quality-oriented network can be implemented further, increasing the user satisfaction with the service quality. The paper is structured as follows. First, the state-of-the-art is discussed in Section 2 to establish a knowledge base that was a starting point in our research. Then, the experiment we used to confirm our theories is discussed in Section 3 as well as the technical background of the measurements. The experimental results are discussed in Section 4 that is followed by Section 5, which concludes the paper.

## 2. Related Work

The idea of QoS measurements is not new; in fact, as soon as the packet switched data networks began to be used for real-time services (audio, video) that were not designed for, the problems with low customer satisfaction arose [4]. For this purpose, the approaches originally designed for public switched telephone networks have been adapted and used. For instance, ITU-T P.800 ACR [5] can be used. However, this method, since it is a part of the subjective tests group, requires human subjects to evaluate the quality, which makes the measurement time-consuming, costly, and susceptible to the mood swings of the assessors.

For these reasons, objective methodologies based on an algorithmic evaluation of speech and video quality have been developed. Among others, Perceptual Evaluation of Speech Quality (PESQ [6,7,8]) and Perceptual Objective Listening Quality Assessment (POLQA [9]) are considered de facto industry standards for speech quality measurement. Both of these standards produce results highly correlated with subjective methods, but on some occasions, the resulting evaluation can be lower due to the higher sensitivity of these algorithms to short-term ripples and peaks. The main drawback of these methods is the fact that they require both original and degraded (captured after transmission) audio signals to work.

This led to the development of methods for speech quality estimation that are trying to deduce the speech quality only from the degraded sound sample or from the parameters of the network transmission. While the former methods are often called non-intrusive, the latter methods are described as parametric. The most commonly used method, which is described as a member of non-intrusive speech quality estimation methodologies (i.e., [10]), is the E-model defined in the ITU-T G.107 standard [11]. This model is based on the fact that the impacts of individual impairments sum up. For local environments and high-quality transit lines (low latency, no echo, etc.), the E-model can be simplified by using some default values as defined in ITU-T G.113 [12].

Moreover, new protocols with new metrics and utility functions are provided in the literature, which are capable of ensuring a QoS level for VoIP calls over different network topologies based on E-model standard such as [13].

The reason that makes the efficient implementation and usage of E-model uneasy comes from the fact that for the E-model to work properly, the network statistics of each individual audio stream need to be collected and processed [14,15]. The data for the packet loss calculation, the used codec, and the delay (if any) can be obtained only in this way. This poses a problem especially for the engineers and developers that need to create specialized probes that would collect the statistics from the mirrored traffic or modify the code of the Public Exchanges (PBX) to do it [16,17]. None of the mentioned is an efficient way, especially because of the limited possibilities to react to the decline of overall speech quality degradation. However, with the advent of SDNs and OpenFlow protocol, the statistics collection can be done directly on the network elements such as OpenFlow switches, from where the data can easily be accessed. A survey of possible ways for statistics collection has been presented in [18], where they use Mininet-based simulation to prove that the statistics about the traffic going through the OpenFlow switch can be collected and that the results obtained highly correlate with the real traffic statistics.

Scientific papers [19,20] have also been published that focus directly on the use of SDN controller and OpenFlow protocol to improve speech quality in VoIP traffic, but both work with knowledge of ports that are used by IP telephony protocols to transmit signaling or voice traffic. Therefore, there is no real measurement of speech quality using objective methods. In addition, a tool called OpenNetMon has been presented in [21]. The tool allows monitoring packet loss, latency, and throughput in the SDN networks based on the OpenFlow protocol, and this information can then be used for the purposes of traffic engineering to enhance the overall QoS.

Although as mentioned above, there are several approaches to use the OpenFlow switches as the network probes to get the traffic flow statistics, the research gap that has been identified by the authors lies with the fact that no real technical solution to measure the speech quality in the OpenFlow-based SDNs has been developed yet, and hence, no comparison with the objective methods for speech quality measurements has been performed either. This paper tries to fill this gap.

## 3. Experimental Setup

In this section, we describe the experimental setup we have used to implement the QoS statistics measurements in the SDN networks. The collection of data is based on the findings described in [18,21], but it is enhanced and further refined by adding speech quality estimation capabilities and reactive mechanisms that allow changing the routing of the media stream in the network, which is based on the calculation of the speech quality estimate.

### 3.1. Simulation Environment

The simulations we performed were conducted using the Mininet simulation tool that is widely used in the SDN community. We used a server in a virtual environment that had the following hardware and software specifications:8C/16T CPU@2400 MHz,16GB DDR4 RAM@2933 MHz,Debian 10 x64,Mininet 2.2.2,Open vSwitch 2.10.1,Ryu Controller 4.30,SIPp 3.5,Asterisk 13.22.

Ryu controller was linked to three instances of Open vSwitch that were interconnected in square-shaped topology. At one end, the SIP call generator was realized by SIPp, and at the other end of the topology, there was Asterisk PBX with endpoints configured and RTP ports restricted to range UDP/10000-10500. The complete topology is depicted in Figure 1.

### 3.2. Simulation Procedure

In the test environment, the simulation procedure is based on a sequential generation of SIP calls using the SIPp call generator. The calls are routed to the Asterisk PBX server via the virtual switches using the bridged interfaces. The SIP server (Asterisk) is preconfigured with 500 valid SIP accounts using the PJSIP stack. Moreover, there are also 500 predefined extensions (numbers to call to) that simulate the other party of the call in a way that they automatically answer the call and place the open call channel to an echo application. This application takes all the input audio data and sends it back to the originator, so both directions of the call contain the same audio data. This functionality is one of the basics in the Asterisk configuration, and any supported version of this SIP server can be used to reproduce the experiment. As a way to define the calls, we use a simple approach and preconfigured Asterisk PBX so it opens only the ports in range UDP/10,000–10,500 for the incoming audio. The RTP traffic is defined as symmetric, meaning that an Asterisk-originated RTP stream is sent via the same port. This way, an easy scheme for audio detection is created. This scheme is versatile and can be expanded to the much higher number of ports since there is no additional overhead related to the number of distinct flows/ports.

A similar functionality as described above can be achieved by using the meter tables and “coloring” the packets by changing the information in the IP DSCP field. However, this approach is overly complex, because each SIP server has its port ranges defined in the configuration.

Now, with the knowledge of the configuration of the SIP server, we can instruct the controller to monitor the specified flows. Each time an RTP stream is negotiated and initiated, the switch needs to poll the controller for the instructions on how to process the incoming packets (PACKET_IN OpenFlow message). Based on the destination/source port number on the SIP server side, the controller can easily determine that the communication being established is an RTP packet stream. There is a slight exception to this, because in the given range, RTCP packets can appear as well, but these can be easily distinguished and eliminated based on the Session-layer information that can be extracted on the controller side, or by finding out that the transmission rate of these packets is much lower (units of seconds for RTCP and milliseconds for RTP).

Each time the PACKET_IN OpenFlow message comes to the controller and the port check returns that the communication is to be done on the designated media ports, Ryu Controller inserts the flow description (flow ID, source port, destination port, initial time) to its database to keep track of the flows that it needs to monitor. Then, the controller sends the FLOW_MOD message to insert the rules (match fields, instructions) to the switch to handle the traffic without a need to keep polling the controller. The key factor in this stage is the low no activity timeout value (1 s ≈ 50 packets of RTP in case of G.711 [22]) for each of the flow table entries that results in rapid response to the call termination, about which the controller is informed using the FLOW_REMOVED message. 

As soon as there is an entry in the controller’s database that holds the information about the media flow, the controller starts polling the particular switch (all switches in the path, actually) for the information about the particular flow’s statistics, namely packet count and byte count. This is achieved by using MULTIPART_REQUEST and MULTIPART_RESPONSE messages that are the standardized carrier messages for requesting the statistics information and replaced FLOW_STATS messages in the previous versions of OpenFlow.

The controller polls the switch synchronously every 2 s (a tunable value). By using the values of a byte count and packet count and the fact that the controller knows exactly when the flow was established, the controller can calculate the expected number of packets and subtract the number of packets that are in the statistics to get the difference that tells the controller how many packets have been lost up to this point.

The expected number of packets is dependent on the used codec (packetization, packet size). Hence, for this purpose, two approaches can be taken. First, the controller can learn the codec from the SIP messages. However, this implies that the controller needs to hold the entire SIP logic, which complicates the implementation. In addition, the controller needs to be aware of all the SIP messages; hence, it must be polled each time the SIP message goes through the switch. This would prevent the solution to be scaled to higher-load environments, since the traffic imposed on the controller would cause a massive overhead. Second, it can be guessed by the actual values of packet and byte counts, since each of the codec profiles has an established and well-known setting for these features (i.e., G.711 uses packetization of 20 ms, and the payload is worth 160 B). This “codec guessing” is getting more precise with the increasing length of the call, and by using separate information about all 2-s long segments of communication, even the effects of packet loss concealment (PLC) techniques can be mitigated and the correct codec can be guessed.

Based on the knowledge gained up to this point, the predicted rating R is calculated using the straightforward Equation (1) coming from E-model:(1)$$R=R_{O}-I_{S}-I_{D}-I_{E-EFF}+A$$
where $R_{O}$ is the highest possible value or rating R if no impairment occurs on the path and it is equal to 94.7688 [11]. $I_{S}$ is the impairment caused by the effects that happen simultaneously with the call (i.e., background noise), which defaults to 1.4136. $I_{D}$ is the impairment caused by echo end delay, and in most cases, it can be omitted if the delay is not higher than 150 ms. $I_{E-EFF}$ is the effective equipment impairment factor (caused by a codec in our case), and it is calculated using the standard formula defined in [11]. The values for its calculation are either taken from the standard if the standard defines them for the given codec. If it does not, then the values are estimated using the regression models that utilize the reference values coming from the quality measurements done by the PESQ algorithm. A, as the advantage factor, is again omitted.

Based on the rating R, the MOS value is calculated again according to [11].

If the MOS value is lower than 2.5 (a tunable value), then the controller is tasked to change the route for the given flow using the FLOW_MOD OpenFlow message.

The delay effects, as is aforementioned, are omitted, but they can be measured as well if needed. However, this measurement cannot come directly from the statistics that OpenFlow switches collect, since OpenFlow defines no facilities for this. However, by using the crafted packets that are inserted into the flow and that carry a timestamp as well, the measurement can be done. This approach was described in [21].

In the following section, the comparison of the results coming from the described method and the ones coming from the PESQ are compared for validity. The injected packet loss is modeled using the 4-state loss model [23], which is the further enhancement of a widely used Gilbert–Elliot model [24]. The 4-state model has four states:State “1”—network is operational and packets are transmitted without any error,State “3”—network is not operational and packets are lost,State “4”—network is operational and packets are lost in an independent fashion, little to no burty losses occur,State “2”—network is not operational and packets are transmitted in an almost independent fashion.

The 4-state model is fully described by the transition probabilities between individual states. In this article, the 4-state model was used in the form defined by Clark in [23] without any modifications. As a loss generator, a Netem emulator is used. The codec used for RTP communication is G.711 [22].

In case of a need for a fine-grained quality estimation, an additional approach can be implemented using the actual voice data, since the switch can be instructed to send each and every single packet to the controller with the voice data. Then, this voice representation can be used as an input to the (convolutional) neural network (CNN) or any other suitable tool. The CNN represents an important supervised learning tool used in different fields such as video and audio analysis, natural language processing, and image recognition [25].

In our experiment, we have implemented a “stub” network coming from the topology used in DeepVocoder [26]. Since the autoencoder employed in the article proved to be efficient in speech compression, we used the modified version of it consisting only of three inception [27] blocks (see Figure 2) to estimate the speech quality from the raw audio data with high precision. The idea behind this approach is that the detected data loss is in terms of signal substituted by the very low-frequency signal (300 Hz) of exactly half the magnitude, which is easily detectable by the convolution filters, and when labeled correctly, the network learns this signal as being the representative of a low speech quality event.

The training data for CNN were constituted from the high-quality recording of political speeches downloaded from YouTube. These data comprised approximately 50% of silence, were in German and English, and were downsampled to narrow-band speech. The overall length of the corpus was 1 h and 57 min, with 80% of randomly selected data being used as a training set and 20% being used as a testing one.

The particular implementation of the CNN, the structure of which is depicted in Figure 2, was based on the following Python tools and libraries:TensorFlow 1.12 with GPU support,Keras 2.2.5,Scikit-learn 0.21.3,Numpy1.17.0.

The CNN model was not run directly in the test environment due to the low computational power of the server used. Instead, a separate computer optimized for CNN-related computations and equipped with two nVidia RTX 2080Ti GPUs was used, and data were fed from the controller to this computer using standardized application calls as depicted in Figure 1.

However, this CNN approach introduces high traffic to the controller and is therefore not suitable and viable to all installments, i.e., complex multielement voice networks in single-controller domain, where the excessive statistics polling would make the controller a performance bottleneck of the network.

## 4. Results

First, the 1% packet loss simulation was performed. The 1% threshold is a result of the combination of transition probabilities of the 4-state model and was chosen as an empirically obtained value that gives a good approximation of network behavior with reasonably long periods of both independent and dependent losses. Under this condition, the quality of experience is impaired significantly (MOS decrease of more than 1 point) for short periods of time, while the most of the speech can still be well heard and understood. This way, a reasonably balanced dataset for CNN. The specific values of transition probabilities of 4-state model are of little importance for this article and can be fine-tuned based on the application needs as long as both random and bursty losses occur in the simulation. However, for the sake of experiment reproducibility, they were set as follows:p13 = 0.005,p31 = 0.99,p32 = 0.005,p23 = 0.9,p14 = 0.005.

In Figure 3, the evolution of the overall cumulative loss is depicted showing the gradual convergence of the model to the selected and specified value of 1%. The spikes in the chart represent a loss event with either individual and separated losses or the bursty loss period that can be distinguished by the relative height of the spike.

With this setup and the loss events generated as described, the three models used to estimate the speech quality were employed. First, the basic E-model calculation takes into account only the packet loss and the delay that is longer than the simulated jitter buffer (100 ms). Second, PESQ calculation using the reference implementation of PESQ compares the original and degraded speech. Both of these work on the sliding time window of 5 seconds to achieve reasonable resolution (100 packets). Third, the CNN-based model works with just the same window as the jitter buffer. This is made possible by the fact that CNN does not compare the signals in any way and only tries to find the features related to natural speech.

Figure 3 depicts the speech quality estimation provided by these three models as well.

In Figure 3, we can see that both PESQ (MOS_LQO) and CNN-based (MOS_LQE[NN]) speech quality calculations tend to superimpose a sort of white noise. This is caused by the fact that the quality estimation done by these methods takes the speech signal into account as well-meaning that the silence periods can cause a difference in the overall result. It is also clear that given the fact that the CNN-based model uses just a small 100 ms long window, it is capable of achieving better result granularity than any of the other two methods. On the other hand, we can see that even a low packet loss (i.e., one packet in five) is quickly classified as a major degradation of speech quality (MOS of around 1), which may cause problems in some environments, and hence, a longer window length should be considered based on the expected results. However, for the sake of this article, we can see that the CNN-based method that takes the data from the SDN controller can be used to determine the speech quality with sufficient accuracy that is required for monitoring purposes.

As it is evident from the Figure 4 and Figure 5, the modeled quality of speech preserves a high correlation with PESQ results and detects the loss events correctly. In Figure 4, the first subfigure shows the evolution of a cumulative packet loss. The system was set to simulate the overall packet loss of 3% (orange dashed line), and the measured cumulative packet loss eventually fluctuates around this threshold. The peaks in the subfigure describe packet loss events, and as the time basis grows, the significance (or magnitude) of these peaks decreases. Further subfigures show the result of MOS measurement using E-Models PESQ and CNN respectively, where CNN again tends to penalize bursty losses due to the shorter time frame. The overall correlation of CNN with PESQ is clear; however, the occurrence of local minima in CNN-based model grows higher in areas where the PESQ result is lower.

The same conclusion can be obtained from Figure 5 depicting the situation when the simulation environment targets the overall losses to 5% threshold.

The respective transition probabilities in these cases were specified as follows:p13 = 0.005,p31 = 0.99,p32 = 0.005,p23 = 0.9,p14 = 0.005,

for 3% loss and
p13 = 0.005,p31 = 0.99,p32 = 0.005,p23 = 0.9,p14 = 0.005,

for 5% loss.

From the simulated loss events and the traffic going through the network, it is clear that the convergence of the CNN model was deep enough to successfully generalize the impact of different types of packet losses caused potentially by both network loss events and excessive delays. To illustrate the rate of convergence, Figure 6 shows the MSE error for the specified network as it was reported during training and testing. Each experiment was performed five times (individual colors of dots in the charts), and then, the mean was calculated and plotted as a green line.

The figure shows rapid and stable convergence to near-zero error rate for both data sets without any significant artifacts. This leads to the conclusion that the CNN-based model not only successfully learned the relation between input (packet loss) and output (MOS) data for the data it has been learning from, but it was also able to generalize the calculation to successfully estimate the output quality for the data it has never seen. The repetition of the experiment also verified that the convergence of the model was not a random event.

## 5. Conclusions

In this paper, a novel approach to how to make a speech quality measurement probe from the standard OpenFlow switch was presented. This approach calculates the statistics of each individual RTP stream/flow based on the information that is read by the controller from the flow tables that are inherently present in these switches. This way, an easily implementable measurement of the call quality on the per call basis is created. The method also allows for a rapid response in case of call quality drops, meaning that the routing of the calls can be changed dynamically on the network levels.

This method uses the information about the codec and the measured packet loss. The delay characteristics are added using the packet injection approach where artificially created packets are introduced to the traffic and the round trip time is measured based on the ICMP response.

In cases where a good quality estimation is required and increased traffic overhead is not an issue, an additional layer of analysis can be added consisting of a convolutional neural network. This network takes input data in the form of a synchronized bitstream with zeros added to the place of missing packets and returns the value of an estimated MOS score. While this approach builds upon samples analyzed by PESQ, it allows for improved accuracy due to the possible incorporation of delay characteristics. Unlike the E-model, which is a standard approach when it comes to online speech quality monitoring, this model improves the overall accuracy by learning from PESQ-analyzed samples. This way, a hybrid solution with the best features of both worlds can be implemented.

As future work, we plan to refine the codec guessing system so it can work in secured TLS environments as well. On top of that, we plan to optimize the way in which the statistics are read from the switches so that the polling of multiple flows can be aggregated into the single message exchange.

## Figures and Tables

**Figure 1 sensors-21-03477-f001:**
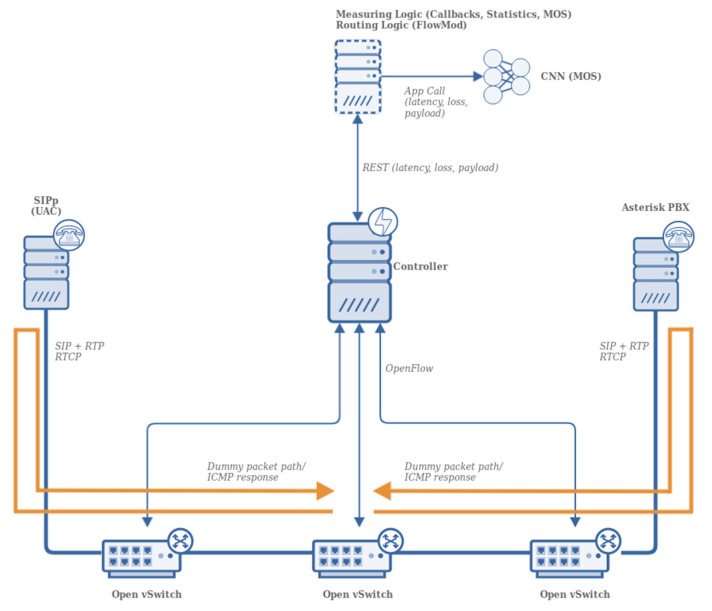
Logical test topology. SIPp generates calls that are routed through OpenFlow switches. Each time a new flow arrives at a switch, the controller is informed using OpenFlow.

**Figure 2 sensors-21-03477-f002:**
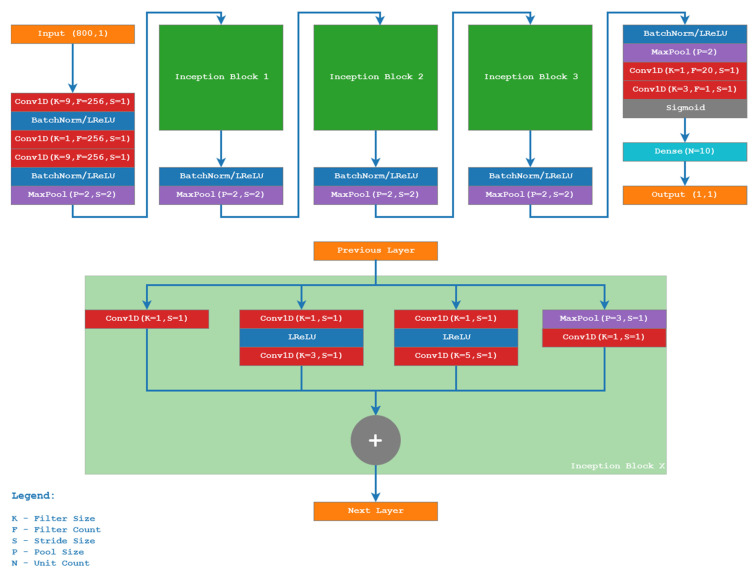
Structure of the CNN based on the Inception model.

**Figure 3 sensors-21-03477-f003:**
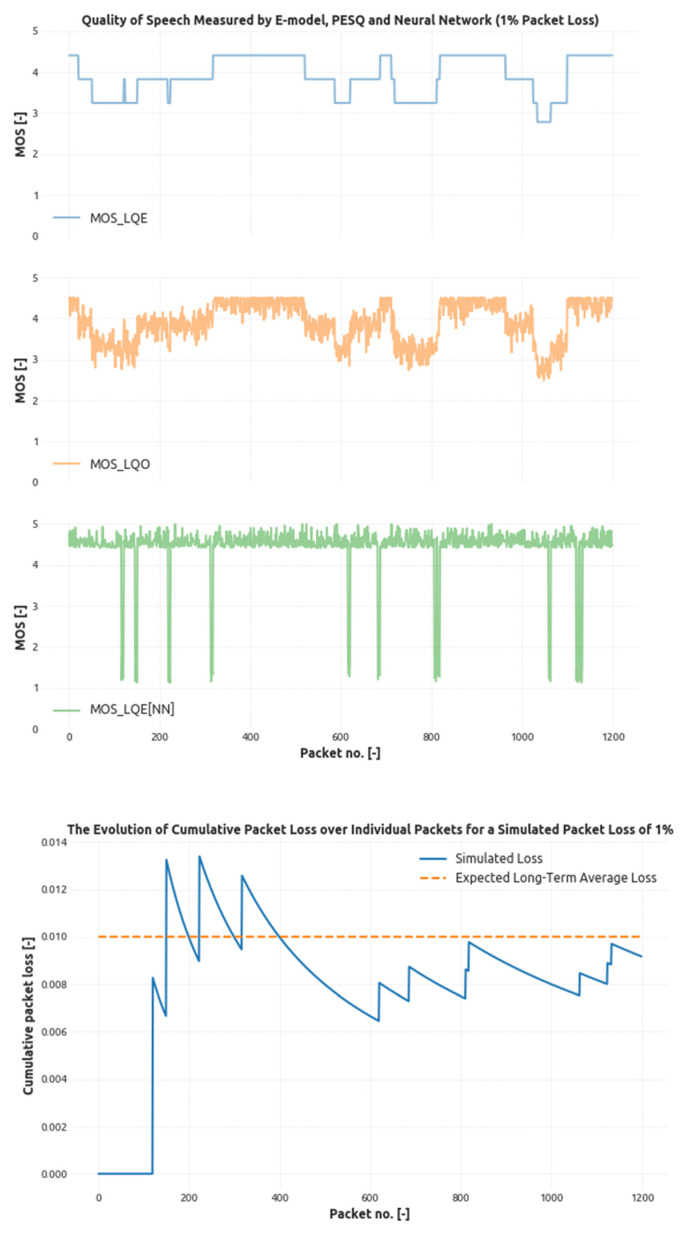
The evolution of the cumulative packet loss generated by the four-state loss model with given probabilities targeting the 1% loss threshold and related speech quality estimated by E-model, PESQ, and CNN.

**Figure 4 sensors-21-03477-f004:**
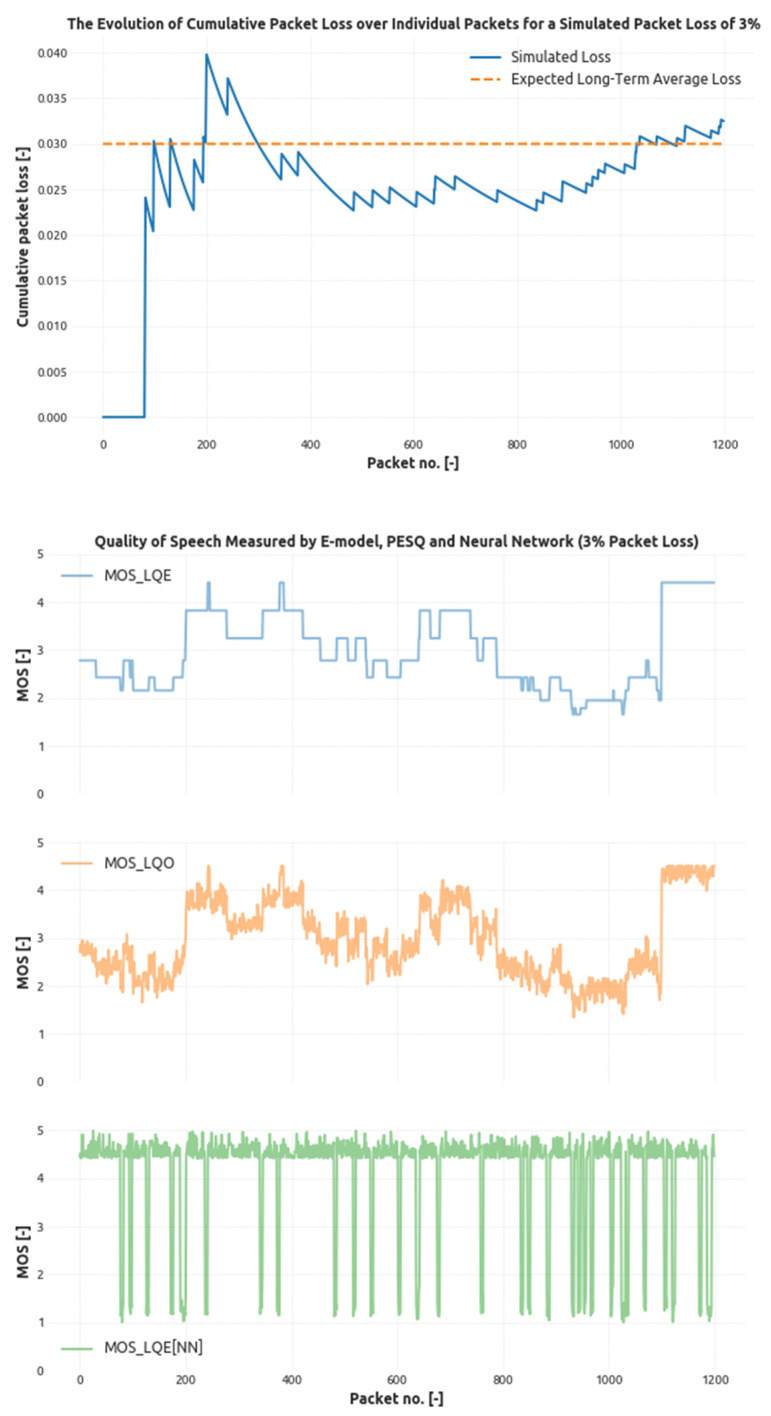
The evolution of the cumulative packet loss generated by the four-state loss model with given probabilities targeting the 3% loss threshold and related modeled speech quality score.

**Figure 5 sensors-21-03477-f005:**
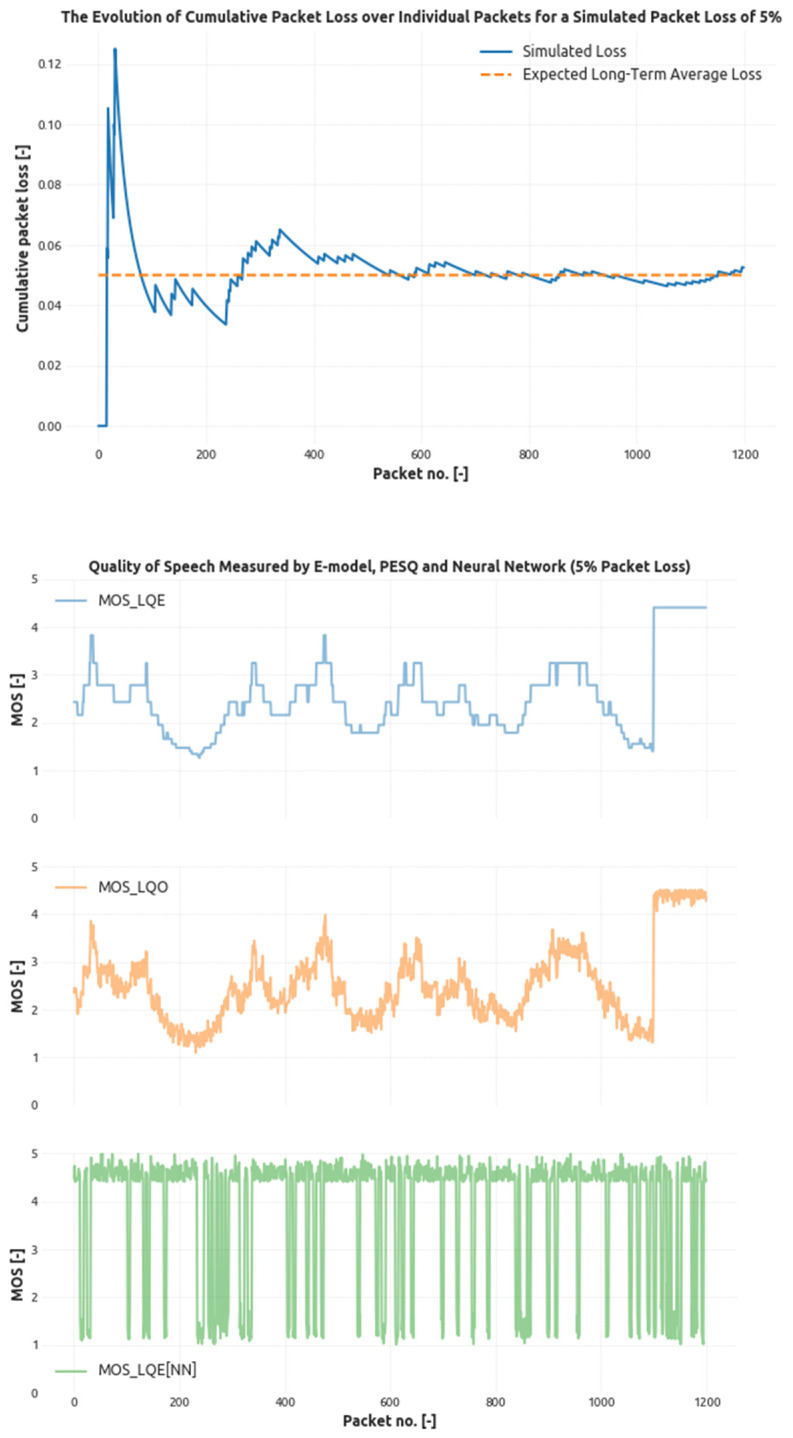
The evolution of the cumulative packet loss generated by the four-state loss model with given probabilities targeting the 5% loss threshold and related modeled speech quality score.

**Figure 6 sensors-21-03477-f006:**
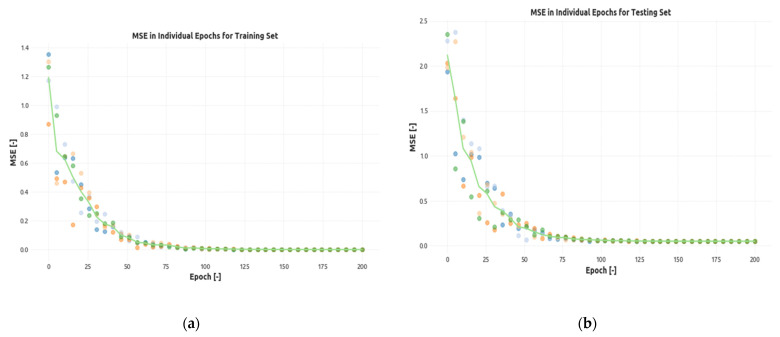
The convergence of the neural network model for both (**a**) training and (**b**) test sets (each experiment was performed five times—individual colors of dots in the charts).

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
