# Peer review of "Augmenting Speech Quality Estimation in Software-Defined Networking Using Machine Learning Algorithms"

_sensors, 2021, doi:10.3390/s21103477_

Round 1
Reviewer 1 Report
The authors present a work that represents an interesting contribution related to the estimation of the quality of service in SDN,through the use of automatic learning tools. The references are appropriate to the context and the work is promising when using tools that coexist within the SDN stack. I suggest the following considerations to improve the quality of work: 1.- Although the work is related to the use of CNN convolutional neural networks, there is not enough evidence of their use.
I suggest abounding within the implementation, as well as the tools and topologies used to implement them. 2. The discussion of Figures 3, 4, 5 and 6 should be clearer to improve their understanding.
Author Response
Comment 1: The authors present a work that represents an interesting contribution related to the estimation of the quality of service in SDN, through the use of automatic learning tools.
Comment 2: The references are appropriate to the context and the work is promising when using tools that coexist within the SDN stack.
Response 1: Thank you very much for your appreciation and comments.
Comment 3: Although the work is related to the use of CNN convolutional neural networks, there is not enough evidence of their use.
Response 2: Thank you for your comment. From our perspective, the CNN model we used was described thoroughly (structure – fig. 2, training and test data selection and features – sec. 3.2 and the performance of the CNN in terms of MSE for both these sets – fig.6) with only implementation details missing. As the results should not be dependent on the particular implementation these were omitted for the paper, but we acknowledge the fact that this missing piece of information may cause doubts, so we added two paragraphs at the end of sec. 3.2.
Comment 4: I suggest abounding within the implementation, as well as the tools and topologies used to implement them. 2. The discussion of Figures 3, 4, 5 and 6 should be clearer to improve their understanding.
Response 3: Thank you for your comment. As a response to the previous comment we enhanced the description of the testbed by adding the CNN-related implementation details. We also extended several paragraphs in sec. 4 in an effort to make the explanation of the Figs 3,4,5 and 6 clearer.
Reviewer 2 Report
1. Most of the proposed experimental methods refer to previous works such as [11, 19, 29, 22, 23, 24].
However, it is needed to explain what the authors' ideas differ from these previous works.
2. The authors shall explain in detail how the introduced CNN technique improves the efficiency of traffic quality estimation.
3. The authors shall specify the terminology description below.
* In Line 275, the authors shall define 'threshold' and how it computed or acquired
* In Line 276, the authors shall provide definition and explanation for '4 state model'
4. It is needed to increase the resolution of Figure 4, 5 and 6.
Author Response
Comment 1: Most of the proposed experimental methods refer to previous works such as [11, 19, 29, 22, 23, 24]. However, it is needed to explain what the authors' ideas differ from these previous works.
Response 1: Thank you for your comment. Work in [11] describes a computational model that we use as a basis for the CNN-based model. The rest of the references points to the individual aspects of employing OpenFlow to network monitoring, Markov chain packet loss models, CNNs in image classification and CNN-based VoCoder (author`s article). In this article, we put the individual pieces together to make a Voice Quality Estimation tool based on OpenFlow switches that can utilize a CNN-based model to estimate the quality of call. Unlike PESQ/POLQA this approach does not need a license and can incorporate delay characteristics as well. And unlike E-model the resulting accuracy is much closer to PESQ. This idea has been added to both abstract and conclusion.
Comment 2: The authors shall explain in detail how the introduced CNN technique improves the efficiency of traffic quality estimation .
Response 2: Thank you for your comment. The modifications made as a response to previous comment should address this one too. In general, CNN-based model allows for online monitoring while reaching near-PESQ accuracy.
Comment 3: The authors shall specify the terminology description below.
* In Line 275, the authors shall define 'threshold' and how it computed or acquired
* In Line 276, the authors shall provide definition and explanation for '4 state model'
Response 3: Thank you for your comment. We acknowledge the fact that the percentual thresholds were not explained in a sufficient manner. We have added a more detailed description to the sec. 4 in an effort to make it more understandable. As for the 4-state model, the description of the model and better reference to literature was added to sec. 3.2.
Comment 4: It is needed to increase the resolution of Figure 4, 5 and 6.
Response4: Thank you for your comment. The resolution of the mentioned figures has been increased and changes were made directly in the manuscript.
Reviewer 3 Report
Authors claim to propose an experimental setup to estimate the quality of speech according to the statistics collected by SDN controllers. Some related works are described and an attempt to identify their drawbacks was made.
The layout of the paper is adequate and balanced between what is related work and what is the contribution with the paper.
Despite the relevance of the topics and the fact that real scenarios are easily identified, the paper has some major flaws and a substantial reformulation should be made, in the case of the paper is accepted.
1) A proof reading should be made, mainly in rephrasing some paragraphs and sentences. For example, the introduction is confused and does not clarify what are the aims, the contributions and the major results accomplished.
2) The bibliographic list is somewhat outdated. Some recent references regarding the measuring of estimation of speech in SDN-based networks should be included. For example, you compare your approach which the findings described in [18,19] which are of 2014 and 2015 (line 148). More recent findings in the subject exists and are available. Others way to solve the problem, besides SDN-based approach, could also be interesting.
3 - Abstract:
Abstract is oriented towards a "discussion" and not necessarily a "finding" or a "contribution". The reader becomes confused whether this is a positional paper or a contribution to the field.
Regarding the experimental setup, what can be summarized in the abstract about the findings and the results obtained with this paper?
2 - Introduction:
A general description of the problem is made, but a more in-depth detail about the contributions should be made. Lines 81-83 are vague and should be detailed. Only individual audio calls are used, or also video content? Despite the description of the aims of FORCES, RCP, NETCONF, and openflow protocols, what are the real added value of the SDN and why it is valuable to enhance the measure of audio and video calls? Lines 54-65 try to describe the advantages of SDN, but fall into trivial descriptions and not necessarily in the specific purpose of your paper.
I must say that the introduction should be rewritten to accomplish these issues.
Some typos:
- Missing "(" somewhere (lines 38,39).
- "With first attempts to move from standard decentralized way of network resources management being FORCES in 2003 [1] provides an overview of this technology and its possible nove use cases) or Routing Control Platform (RCP [2])...". This phrase is hard to read and should be rewritten.
3 - Related work:
The concerns addressed previously regarding the need to include up-to-date references.
Lines 139-144 you mention that "no real technical solution to measure the speech quality in the OpenFlow-based SDNs has been developed yet...". Sure? Based on which biblio references? A quick look at the bibliography reveals some related works:
- https://www.sciencedirect.com/science/article/pii/S1389128614002254?casa_token=ToeDYDYpruwAAAAA:F7rWzW9HHqEYHEdKoYE_WNHKfKKV9kvFybejwlKhoJX__MXFV5orkbzkzG-IkCu1iRJlPKZmcw
- https://www.spiedigitallibrary.org/conference-proceedings-of-spie/11018/110181M/Measuring-and-monitoring-the-QoS-and-QoE-in-software-defined/10.1117/12.2518838.short?SSO=1
- https://dl.acm.org/doi/abs/10.1145/3293614.3293651?casa_token=hCaKHh4g1TgAAAAA:tXOlx7igmtJQsLbZVYzrMURvu814CDG9ioH6bAWEB1a7AlBDbGU8S080rwe3uRUQFBqZx77dUgE5
This paragraph should be better written and has to be justified and supported in the literature, as it represents the justification for your contributions in this paper.
4) Experimental setup:
- The references in which you based your collected data are [18] [19], that dated from 2014 and 2015. Are there any updates to these biblio references?
- The simulated environment seems to be adequate to the experiments. I'd complement the setup explanation with the software versions and a description of the VMs that implements Open vSwitch (CPU, memory, etc...).
- Figure 2 is not pointed out in the text.
- Simulation procedure (section 3.2) deserves a schematic explanation to guide the reader.
- What is the meaning of Figure 2? Is it the CNN architecture? It should be described and framed in the simulation procedure.
- The "predicting rating" (R) is related to what? Too dense this part of the text and fall by parachute in the text. It should be described according to CNN architecture.
- Lines 272-273: why? This sentence should be better explained. CNN is the "machine learning side" of your paper and is a key component of your architecture. Apparently it represents a "problem" regarding its implementation in real scenarios. Could you clarify?
- It is not understandable the overall pipeline, that is how data is preprocessed and feed the CNN.
- Globally, section 3.2 is too dense and deserves to be better explained.
5) Results
- Which results you are obtaining in each experiment?
- You mention that "The 1% threshold is a result of 275 the combination of transition probabilities of the 4-state model." (lines 275-276). What is this? Could you better explain why 1% and 5% thresholds? And what is the "s-state-model"?
- What are p13, p31... (lines 280-284)? Packet loss?
- Lines 289-297 are duplicated!
- Lines 299-306 - Is this a "fair" comparison between the three models? The same input data (videos and audios) were applied to the same models?
- How MOS was calculated? According to [11]? I can not see where the calculation is described in [11]. Could you elucidate?
- A more suitable caption should be put in Figure 3. It seems that the fourth plot is related with the cumulative packet loss and not with MOS. But are included in the same picture. A better data visualization should be applied in the figures.
6) Conclusions.
- Text should be extended a bit more, in order to better detail the finding observed.
- Any clue about how to deal with TLS traffic? As you work with flows, it is important to go deep with the payload? If so, how do you manage to deal with encrypted data?
Summing up, the subject treated in this paper is worth investigating and the research has merit. However, the paper has several flaws and should be rewritten.
Author Response
Comment 1: Authors claim to propose an experimental setup to estimate the quality of speech according to the statistics collected by SDN controllers. Some related works are described and an attempt to identify their drawbacks was made.
The layout of the paper is adequate and balanced between what is related work and what is the contribution with the paper.
Response 1: Thank you very much for your appreciation and comments.
Comment 2: A proof reading should be made, mainly in rephrasing some paragraphs and sentences. For example, the introduction is confused and does not clarify what are the aims, the contributions and the major results accomplished.
Response 2: Thank you for your comment. The authors performed a detailed proof reading of the manuscript and grammatical adjustments were made directly into the article, where they are also highlighted.
Comment 3: The bibliographic list is somewhat outdated. Some recent references regarding the measuring of estimation of speech in SDN-based networks should be included. For example, you compare your approach which the findings described in [18,19] which are of 2014 and 2015 (line 148). More recent findings in the subject exists and are available. Others way to solve the problem, besides SDN-based approach, could also be interesting.
Response 3: Thank you for your comment. Reference number 18 has been modified to cite a scientific publication which represents the current state of the problematics (see below). Reference number 19 is by far the only scientific publication that focuses directly on the issue of OpenNetMon, and also related scientific works often refer to 19.
- Queiroz, W., Capretz, M.A.M., Dantas, M. An approach for SDN traffic monitoring based on big data techniques, Journal of Netwrok and computer applications, Volume 131, https://doi.org/10.1016/j.jnca.2019.01.016, 2019, pp. 28-39.
Comment 4: Abstract is oriented towards a "discussion" and not necessarily a "finding" or a "contribution". The reader becomes confused whether this is a positional paper or a contribution to the field. Regarding the experimental setup, what can be summarized in the abstract about the findings and the results obtained with this paper?
Response 4: Thank you for your comment. The abstract was modified to reflect the results obtained and summarize the key benefits of the proposed solution.
Abstract: With the increased number of Software-Defined Networking (SDN) installations the datacenters of large service providers are becoming more and more agile in terms of network performance efficiency and flexibility. While SDN is an active and obvious trend in a modern data center design the implications and possibilities it carries for the effective and efficient net-work management are not yet fully explored and utilized. With most of the modern Internet traffic consisting of multimedia services and media-rich content sharing, the quality of multimedia communications is at the center of attention of many companies and research groups. Since SDN-enabled switches have an inherent feature of monitoring the flow statistics in terms of packets and bytes transmitted/lost, these devices can be utilized to monitor the essential statistics of the multimedia communications allowing the provider to act in case of network failing to deliver the required service quality. The internal packet processing in the SDN switch enables the SDN controller to fetch the statistical information of the particular packet flow using the PacketIn and Multipart messages. This information, if preprocessed properly, can be used to estimate higher layer interpretation of the link quality and thus allowing to relate the provided quality of service (QoS) to the quality of user experience (QoE). This article discusses the experimental setup that can be used to estimate the quality of speech communication based on the in-formation provided by the SDN controller. To achieve higher accuracy of the result, latency characteristics are added based on the exploiting of the dummy packet injection into the packet stream and/or RTCP packet analysis. The results of the experiment show that this innovative approach calculates the statistics of each individual RTP stream and thus we obtains a method for dynamic measurement of speech quality, where when quality decreases, it is possible to respond quickly by changing routing at the network level for each individual call. To improve the quality of call measurements, a Convolutional Neural Network (CNN) was also implemented., This model is based on two standard approaches to measuring the speech quality – PESQ and E-model. But unlike PESQ/POLQA, CNN-based model can take delay into account and unlike E-model the resulting accuracy is much higher. The possible use of convolutional neu-ral networks as a mean to acquire the application layer quality of speech is discussed as well to further strengthen the background information upon which the decision about the network's performance is taken.
Comment 5: A general description of the problem is made, but a more in-depth detail about the contributions should be made. Lines 81-83 are vague and should be detailed. Only individual audio calls are used, or also video content? Despite the description of the aims of FORCES, RCP, NETCONF, and openflow protocols, what are the real added value of the SDN and why it is valuable to enhance the measure of audio and video calls? Lines 54-65 try to describe the advantages of SDN, but fall into trivial descriptions and not necessarily in the specific purpose of your paper.
I must say that the introduction should be rewritten to accomplish these issues.?
Response 5: Thank you for your comment. We tried to modify the abstract and introduction to accommodate your objections to contribution. Although the method can be modified to incorporate the video quality measurements as well, in this article, however, we discuss only audio calls. As for the SDN, there is no real added value of using SDN. We propose a method that can enhance the service quality in SDN environment further increasing the user satisfaction. SDN is not a tool for us, it is an environment that we aim to improve. The measurement of audio quality can allow for better services and increase the user satisfaction, which we find important in a highly competitive field of telecommunications.
Comment 6: Missing "(" somewhere (lines 38,39).
Response 6: Thank you for your comment. The typographical error was corrected directly in the manuscript.
Comment 7: "With first attempts to move from standard decentralized way of network resources management being FORCES in 2003 [1] provides an overview of this technology and its possible novel use cases) or Routing Control Platform (RCP [2])...". This phrase is hard to read and should be rewritten.
Response 7: Thank you for your comment. Sentences have been modified as part of a revision of the entire Introduction (please see the Response 5).
Comment 8: The concerns addressed previously regarding the need to include up-to-date references.
Lines 139-144 you mention that "no real technical solution to measure the speech quality in the OpenFlow-based SDNs has been developed yet...". Sure? Based on which biblio references? A quick look at the bibliography reveals some related works:
- https://www.sciencedirect.com/science/article/pii/S1389128614002254?casa_token=ToeDYDYpruwAAAAA:F7rWzW9HHqEYHEdKoYE_WNHKfKKV9kvFybejwlKhoJX__MXFV5orkbzkzG-IkCu1iRJlPKZmcw
- https://www.spiedigitallibrary.org/conference-proceedings-of-spie/11018/110181M/Measuring-and-monitoring-the-QoS-and-QoE-in-software-defined/10.1117/12.2518838.short?SSO=1
- https://dl.acm.org/doi/abs/10.1145/3293614.3293651?casa_token=hCaKHh4g1TgAAAAA:tXOlx7igmtJQsLbZVYzrMURvu814CDG9ioH6bAWEB1a7AlBDbGU8S080rwe3uRUQFBqZx77dUgE5
This paragraph should be better written and has to be justified and supported in the literature, as it represents the justification for your contributions in this paper.
Response 8: Thank you for your comment. A paragraph was added describing the work referred to by the reviewer, where it was found that these scientific publications use different approaches to speech quality assessment and optimization using SDN networks than the proposal presented by the authors in this article. In addition, the first publication mentioned by the reviewer does not directly address the measurement of speech quality within SDN networks.
Scientific papers [26], [27] have also been published that focus directly on the use of SDN controller and OpenFlow protocol to improve speech quality in VoIP traffic, but both work with knowledge of ports that are used by IP telephony protocols to transmit signal-ing, or voice traffic. Therefore, there is no real measurement of speech quality using objec-tive methods.
Also, a tool called OpenNetMon has been presented in [19]. The tool allows to moni-tor monitoring packet loss, latency, and throughput in the SDN networks based on the OpenFlow protocol and this information can then be used for the purposes of traffic engi-neering to enhance the overall QoS.
Although as mentioned above there are several approaches to use the OpenFlow switches as the network probes to get the traffic flow statistics, the research gap that has been identified by the authors lies with the fact that no real technical solution to measure the speech quality in the OpenFlow-based SDNs has been developed yet and hence no comparison with the objective methods for speech quality measurements has been per-formed either. This paper tries to fill this gap.
- Vieira, D., Juca, P., Callado, A. A Solution for QoS Provisioning in VoIP Services on theOpenFlow Platform, Proceedings of the Euro American Conference on Telematics and Information Systems (EATIS), 2018, pp.1-5.
- Rozhon, J., Rezac, F., Safarik, J., Gresak, E., Jalowiczor, J. Measuring and monitoring the QoS and QoE in software defined networking environments, SPIE Proceedings - Signal Processing, Sensor/Information Fusion, and Target Recognition XXVIII, 2019.
Comment 9: The references in which you based your collected data are [18] [19], that dated from 2014 and 2015. Are there any updates to these biblio references?
Response 9: Thank you for your comment. This comment has already been corrected in the context of addressing Comment 3 (please see Response 3).
Comment 10: The simulated environment seems to be adequate to the experiments. I'd complement the setup explanation with the software versions and a description of the VMs that implements Open vSwitch (CPU, memory, etc...).
Response 10: Thank you for your comment. An overview of the hardware and software specifications that were used in the experimental setup was added directly to the manuscript.
The simulations we performed were conducted using the Mininet simulation tool that is widely used in the SDN community. We used a server in a virtual environment that had the following hardware and software specifications:
- 8C/16T CPU@2400 MHz,
- 16GB DDR4 RAM@2933 MHz,
- Debian 10 x64,
- Mininet 2.2.2,
- Open vSwitch 2.10.1,
- Ryu Controller 4.30,
- SIPp 3.5,
- Asterisk 13.22.
Comment 11: Figure 2 is not pointed out in the text.
Response 11: Thank you for your comment. The figure is referenced in a text that describes the purpose and involvement of CNN in the whole concept of measuring and evaluating speech quality (Lines 295 – 322).
In our experiment, we have implemented a "stub" network coming from the topology used in DeepVocoder [24]. Since the autoencoder employed in the article proved to be efficient in speech compression, we used the modified version of it consisting only of three inceptions [25] blocks (see Fig. 2) to estimate the speech quality from the raw audio data with a high precision. The idea behind this approach is that the detected data loss is in terms of signal substituted by the very low- frequency signal (300 Hz) of exactly half the magnitude which is easily detectable by the convolution filters and when labeled correctly the network learns this signal as being the representative of a low speech quality event.
The training data for the CNN were constituted from the high- quality recording of political speeches downloaded from Youtube. These data comprised approximately 50% of silence, were in German and English and were downsampled to narrow-band speech. The overall length of the corpus was 1 hour and 57 minutes, with 80% of randomly selected data being used as a training set and 20% as a testing one.
The particular implementation of the CNN, the structure of which is depicted in Fig. 2, was based on the following Python tools and libraries:
- TensorFlow 1.12 with GPU support,
- Keras 2.2.5,
- Scikit-learn 0.21.3,
- Numpy 1.17.0.
The CNN model was not run directly in the test environment due to the low computational power of the server used. Instead, a separate computer optimized for CNN-related computations and equipped with two nVidia RTX 2080Ti GPUs was used and data were fed from the controller to this computer using standardized application calls as depicted in Fig. 1.
This CNN approach, however, introduces a high traffic to the controller and is there-fore not suitable and viable to all installments, i.e. complex multielement voice networks in single-controller domain, where the excessive statistics polling would make the controller a performance bottleneck of the network.
Comment 12: Simulation procedure (section 3.2) deserves a schematic explanation to guide the reader. What is the meaning of Figure 2? Is it the CNN architecture? It should be described and framed in the simulation procedure.
Response 12: Thank you for your comment. The simulation topology is shown in Figure 1, where the communication between the individual elements is also marked. The description of OpenFlow messages, as described in section 3.2, corresponds to the communication between the SND controller and the Open vSwitch network elements. A description of the meaning of Figure 2 is given in the two final paragraphs of Chapter 3.2, and the involvement of CNN in the test topology is shown in Figure 1.
Comment 13: The "predicting rating" (R) is related to what? Too dense this part of the text and fall by parachute in the text. It should be described according to CNN architecture.
Response 13: Thank you for your comment. As the for the rating R, we have slightly modified the sentence hoping to achieve better clarity. This part is not directly related to CNN architecture although due to article layout issues appear together with Fig. 2. The reference to the E-model and its definition is also present.
Comment 14: Lines 272-273: why? This sentence should be better explained. CNN is the "machine learning side" of your paper and is a key component of your architecture. Apparently it represents a "problem" regarding its implementation in real scenarios. Could you clarify?
Response 14: Thank you for your comment. We have added an example, where this setup could pose a threat to the network stability.
This CNN approach, however, introduces a high traffic to the controller and is there-fore not suitable and viable to all installments, i.e. complex multielement voice networks in single-controller domain, where the excessive statistics polling would make the controller a performance bottleneck of the network.
Comment 15: It is not understandable the overall pipeline, that is how data is preprocessed and feed the CNN.
Response 15: Thank you for your comment. The preprocessing is a simple one. We take 100 ms samples of voice data (5 packets, 5x160 bytes = 800 samples – as seen in Fig. 2). In case of packet loss, we zero-pad (also in the article) effectively introducing silence. As to why 100 ms, it is an arbitrary value that can change from installment to installment based on quality expectations, desired sensitivity, “smoothness” of the result, codec used, etc. The article cannot possibly cover all possible scenarios.
Comment 16: Globally, section 3.2 is too dense and deserves to be better explained.
Response 16: Thank you for your comment. As a result of the modifications made based on other reviewers` comments this section has been modified. We believe it describes the situation better now.
Comment 17: Which results you are obtaining in each experiment? You mention that "The 1% threshold is a result of 275 the combination of transition probabilities of the 4-state model." (lines 275-276). What is this? Could you better explain why 1% and 5% thresholds? And what is the "s-state-model"?
Response 17: Thank you for your comment. As a response to other reviewer`s comments we have added a better explanation of the situation and we strongly believe it will accommodate your doubts.
Comment 18: What are p13, p31... (lines 280-284)? Packet loss?
Response 18: Thank you for your comment. As per a response to your previous comment, this part has been changed to make it easier to follow.
Comment 19: Lines 289-297 are duplicated!
Response 19: Thank you for your comment. The typographical error was corrected directly in the manuscript.
Comment 20: Lines 299-306 - Is this a "fair" comparison between the three models? The same input data (videos and audios) were applied to the same models?
Response 20: Thank you for your comment. Yes, same audio data was used in all measurements as per description in sec. 3.2.
Comment 21: How MOS was calculated? According to [11]? I cannot see where the calculation is described in [11]. Could you elucidate?
Response 21: Thank you for your comment. MOS was calculated according to [11], analyzed using PESQ and computed using trained CNN. In the article, there is an equation for rating R that comes from [11]. In [11], there is also an equation that allows for the calculation of MOS from R, please see Annex B on pp. 15. From our articles perspective, the key information comes from R and since the MOS is then calculated in a standardized fashion, we believe reference is sufficient.
Comment 22: A more suitable caption should be put in Figure 3. It seems that the fourth plot is related with the cumulative packet loss and not with MOS. But are included in the same picture. A better data visualization should be applied in the figures.
Response 22: Thank you for your comment. The resolution of the pictures has been increased. The captions should now be clearly visible.
Comment 23: Text should be extended a bit more, in order to better detail the finding observed. Any clue about how to deal with TLS traffic? As you work with flows, it is important to go deep with the payload? If so, how do you manage to deal with encrypted data?
Response 23: Thank you for your comment. This is just a theoretical thing so far, but we have been working on the heuristic model that would estimate the type of the audio present. This, however, will always be a probabilistic not deterministic approach.
Reviewer 4 Report
The paper is interested in an UpToDate topic. The authors offer interesting solution. The aim of the paper is an experimental setup that can be used to estimate the quality of speech communication based on the information provided by the SDN controller. To achieve higher accuracy of the result, latency characteristics the authors added based on the exploiting of the dummy packet injection into the packet stream and/or RTCP packet analysis.
The structure of the paper seems to be proper. They started with Introduction, continued with literature review, methodology, results, results discussion, and conclusion. The range of mentioned related works is not significant. They used a small number of sources. They included a few of their formerly published papers.
Methodology - experiment is set up properly. The results are presented in an understandable way. There are some typing errors. They should be corrected (for example row 327). Conclusion is also all right. However, I suggest defining the contribution of the paper for science and practice.
Overall, the paper is potential and should be published after minor revisions.
I offer just two recommendation to improve the paper:
Firstly, the overview of similar research should be extended.
Secondly, define the contribution of the paper for science and practice.
Author Response
Comment 1: The paper is interested in an UpToDate topic. The authors offer interesting solution. The aim of the paper is an experimental setup that can be used to estimate the quality of speech communication based on the information provided by the SDN controller. To achieve higher accuracy of the result, latency characteristics the authors added based on the exploiting of the dummy packet injection into the packet stream and/or RTCP packet analysis.
Comment 2: The structure of the paper seems to be proper. They started with Introduction, continued with literature review, methodology, results, results discussion, and conclusion. The range of mentioned related works is not significant. They used a small number of sources. They included a few of their formerly published papers.
Comment 3: Overall, the paper is potential and should be published after minor revisions
Response 1: Thank you very much for your appreciation and comments.
Comment 4: Firstly, the overview of similar research should be extended.
Response 4: Thank you for your comment A paragraph has been added describing two scientific papers that directly address the evaluation of speech quality using SDN networks, where it was found that these scientific publications use different approaches to speech quality assessment and optimization using SDN networks than the proposal presented by the authors in this article.
Scientific papers [26], [27] have also been published that focus directly on the use of SDN controller and OpenFlow protocol to improve speech quality in VoIP traffic, but both work with knowledge of ports that are used by IP telephony protocols to transmit signal-ing, or voice traffic. Therefore, there is no real measurement of speech quality using objec-tive methods.
Also, a tool called OpenNetMon has been presented in [19]. The tool allows to moni-tor monitoring packet loss, latency, and throughput in the SDN networks based on the OpenFlow protocol and this information can then be used for the purposes of traffic engi-neering to enhance the overall QoS.
Although as mentioned above there are several approaches to use the OpenFlow switches as the network probes to get the traffic flow statistics, the research gap that has been identified by the authors lies with the fact that no real technical solution to measure the speech quality in the OpenFlow-based SDNs has been developed yet and hence no comparison with the objective methods for speech quality measurements has been per-formed either. This paper tries to fill this gap.
- Vieira, D., Juca, P., Callado, A. A Solution for QoS Provisioning in VoIP Services on theOpenFlow Platform, Proceedings of the Euro American Conference on Telematics and Information Systems (EATIS), 2018, pp.1-5.
- Rozhon, J., Rezac, F., Safarik, J., Gresak, E., Jalowiczor, J. Measuring and monitoring the QoS and QoE in software defined networking environments, SPIE Proceedings - Signal Processing, Sensor/Information Fusion, and Target Recognition XXVIII, 2019.
Comment 5: Secondly, define the contribution of the paper for science and practice.
Response 5: Thank you for your comment. We modified several parts of the article (specifically in the Results and Conclusion chapter) to address the contribution of the paper.
Round 2
Reviewer 1 Report
Suggestions have been addressed.
Reviewer 2 Report
The authors improved the quality of the paper by faithfully reflecting the opinions of the reviewers.
Reviewer 3 Report
Authors have made an adequate review to the reviewer's comments. Some major flaws of the paper were fixed, and some unclear content is now better explained.
Congrats!